# Elevated liver glycogenolysis mediates higher blood glucose during acute exercise in Barth syndrome

George G. Schweitzer[1], Grace L. Ditzenberger[2], Curtis C. Hughey[3], Brian N. Finck[1], Michael R. Martino[1], Christina A. Pacak[4], Barry J. Byrne[5], William Todd Cade[2,6]*

1 Center for Human Nutrition, Washington University School of Medicine, St. Louis, Missouri, United States of America, 2 Doctor of Physical Therapy Division, Duke University School of Medicine, Durham, NC, United States of America, 3 Department of Medicine, Division of Molecular Medicine, University of Minnesota, Minneapolis, MN, United States of America, 4 Department of Neurology, University of Minnesota School of Medicine, Minneapolis, MN, United States of America, 5 Department of Pediatrics, College of Medicine, University of Florida, Gainesville, FL, United States of America, 6 Program in Physical Therapy, Washington University School of Medicine, St. Louis, Missouri, United States of America

* todd.cade@duke.edu

**Data Availability Statement:** All relevant data are within the paper and its Supporting Information files.

## Abstract

Barth syndrome (BTHS) is an X-linked recessive genetic disorder due to mutations in the Tafazzin (TAFAZZIN) gene that lead to cardiac and skeletal muscle mitochondrial dysfunction. Previous studies in humans with BTHS demonstrate that the defects in muscle mitochondrial oxidative metabolism result in an enhanced reliance on anaerobic metabolism during exercise to meet energy demands of muscular work. During exercise, the liver normally increases glucose production via glycogenolysis and gluconeogenesis to match the elevated rate of muscle glucose uptake and meet the ATP requirements of working muscle. However, the impact of Tafazzin deficiency on hepatic glucose production and the pathways contributing to hepatic glucose production during exercise is unknown. Therefore, the purpose of this study was to quantify *in vivo* liver gluconeogenesis and glycogenolysis in Tafazzin knockdown mice at rest and during acute exercise. **METHODS:** Male TAFAZZIN shRNA transgenic (TG) and wild-type (WT) mice completed exhaustive treadmill running protocols to test exercise tolerance. Mice underwent $^2$H- and $^{13}$C-stable isotope infusions at rest and during a 30-minute treadmill running bout to quantify hepatic glucose production and associated nutrient fluxes under sedentary conditions and during acute exercise. Circulating and tissue (skeletal muscle and liver) samples were obtained during and following exercise to assess static metabolite levels. **RESULTS:** TG mice reached exhaustion sooner during exhaustive treadmill running protocols and exhibited higher plasma lactate concentrations after exhaustive exercise compared to WT mice. Arterial glucose levels were comparable between genotypes at rest, but higher in TG mice compared to WT mice during exercise. Consistent with the higher blood glucose, TG mice showed increased endogenous glucose production owing to elevated glycogenolysis compared to WT mice during exercise. Total gluconeogenesis, gluconeogenesis from glycerol, gluconeogenesis from phosphoenolpyruvate, pyruvate cycling, total cataplerosis, and anaplerotic fluxes were similar between TG and WT mice at rest and during exercise. However, lactate dehydrogenase flux and TCA

**Funding:** This study was supported by the Barth Syndrome Foundation, the Vanderbilt University Mouse Metabolic Phenotyping Center for core services (NIDDK, National Institutes of Health (NIH), DK059637), and NIH grants U01 DK094416, P30 DK056341 (Washington University Nutrition Obesity Research Center), P30 DK092950 (Washington University Center for Diabetes Translation Research), P30 DK20579 (Washington University Diabetes Research Center) UL1 TR002345 (Washington University Clinical and Translational Science Award), and R01 DK117657 (Finck).The funders had no role in the study design, data collection, decision to publish or preparation of the manuscript.

**Competing interests:** The authors declared no conflict of interest.

cycle fluxes trended higher in TG mice during exercise. Liver glycogen content in TG was higher in TG vs. controls. **CONCLUSION:** Our data in the Tafazzin knockdown mouse suggest that elevated anaerobic metabolism during rest and exercise previously reported in humans with BTHS are supported by the finding of higher hepatic glycogenolysis.

## Introduction

Barth syndrome (BTHS) is an X-linked recessive genetic disorder resulting from mutations in the gene encoding Tafazzin (*TAFAZZIN*) with individuals presenting with cardiomyopathy, skeletal myopathy, muscle weakness, exercise intolerance, neutropenia, and growth delays [1,2]. Tafazzin is an acyltransferase enzyme that remodels cardiolipin; a phospholipid enriched in the inner mitochondrial membrane that plays important regulatory roles in mitochondrial intermediary metabolism [3]. Mutations in Tafazzin universally result in an accumulation of the unremodeled form of cardiolipin in the mitochondria and frequently lower levels of remodeled cardiolipin, ultimately impacting membrane stabilization and oxidative phosphorylation to synthesize ATP [4,5].

Defects in mitochondrial structure and function lead to exercise intolerance; a hallmark characteristic in the mouse model of BTHS [6] and humans with BTHS [7]. In healthy individuals, oxidation of fatty acids and glucose in mitochondria supplies ATP to meet the high energy demands of muscular work during exercise [8]. However, ATP production from fatty acid oxidation during exercise in BTHS is severely limited and appears to contribute to exercise dysfunction [9]. Previous studies in BTHS suggest that defects in mitochondrial oxidative metabolism result in an enhanced reliance on anaerobic metabolism to meet energy demands, potentially as a means to compensate for impaired mitochondrial aerobic metabolism. For example, increased blood lactate concentrations at lower workloads in BTHS suggest enhanced anaerobic glucose metabolism in BTHS [9]. Since lactate is the end-product of glycolysis, increased blood lactate concentrations may be indicative of enhanced flux through anaerobic pathways, particularly during prolonged exercise bouts. However, glycolysis produces only two molecules of ATP per glucose molecule, which is significantly less than nutrient disposal via mitochondrial oxidative metabolism, and thus quantitatively greater amounts of glucose are needed to meet ATP demands in highly glycolytic states. Interestingly, individuals with BTHS exhibit higher resting insulin sensitivity [10] and greater glucose turnover during rest, moderate exercise, and post-exercise recovery compared to non-affected healthy controls [9].

Substrate use and adaptive mechanisms involved in meeting energy demands in BTHS are incompletely understood. During acute exercise, the liver normally increases glucose production to match the increased rate of muscle glucose uptake required to meet the ATP demands of muscular work [11,12]. The greater glucose production is mediated by hepatic glycogenolysis and by gluconeogenesis wherein intermediates such as glycerol and phosphoenolpyruvate (via pyruvate, lactate, and amino acids) are used to generate glucose *de novo*. The precursors required for gluconeogenesis are supplied partly by the Cori and Cahill cycles in which lactate and alanine, respectively, are released by muscle into the circulation during exercise and are subsequently extracted from the circulation by the liver [13,14]. Recent work using advanced tracer-based approaches and metabolic flux analysis has demonstrated that impaired hepatic gluconeogenesis from phosphoenolpyruvate impedes blood glucose production during acute exercise [15]. However, whether Tafazzin deficiency impacts hepatic glucose production during exercise and which pathway(s) might be impacted in BTHS is unknown. Therefore, the

purpose of this study was to use stable isotopes to quantify liver gluconeogenesis and glycogenolysis in Tafazzin knockdown mice at rest and during acute exercise. We hypothesized that gluconeogenesis and glycogenolysis would be elevated in Tafazzin knock-down mice during moderate intensity exercise compared to wild-type (WT) controls to meet the energetic demands of muscular work.

## Materials and methods

### Generation of Tafazzin Deficient Mice

The Washington University (#20170034) and Vanderbilt University (#M2000050-00) Institutional Animal Care and Use Committees approved all animal studies. Male TAFAZZIN shRNA transgenic (TG) and Wild-Type (WT) adult mice were used for this study. The doxycycline-inducible short hairpin RNA (shRNA)-mediated Tafazzin knockdown model has been previously described [6,16–18]. Briefly, TG mice contained a tetracycline-inducible shRNA that is targeted to knockdown Tafazzin to mimic the effects of loss of function mutations that occur in humans. This model is inducible in that, upon administration of the tetracycline analog doxycycline, the Tet repressor that is blocking shRNA expression is released. At 2 months of age, mice were started on doxycycline (200 mg/kg) chow and continued on the chow for the remainder of the study. Mice were bred at Washington University and transferred to the Vanderbilt University Mouse Metabolic Phenotyping Center for stable isotope infusions.

### Exercise challenge

For initial exercise studies shown in Fig 1, male mice that had been on doxycycline chow for 6 months were run to exhaustion on a closed 6-lane treadmill (Columbus Instruments) equipped with a shock grid at the back of each belt that delivered a mild electrical stimulus to encourage continuous running. Food was removed 18 hours before exercise, water was available ad libitum, and bedding was replaced with aspen chip bedding. The 18 h fast was designed to remove the effect of prior meals on exercise performance and to test our hypothesis that gluconeogenesis would be different. Depleting liver glycogen with an 18 h fast would place greater reliance on gluconeogenesis to possibly enhance differences between genotypes. Mice were acclimated to the treadmill with 0˚ incline at 0 m/min for 5 minutes. The speed was then increased to 5 m/min and maintained for 5 minutes. After 5 minutes of continuous running the speed was increased so that after the first 5 minutes, the speed reached 10 m/min, 5 minutes later the speed reached 15 m/min, etc, until mice reached exhaustion. Speeds did not exceed 30 m/min. Exhaustion was determined by refusal of mice to remain on the treadmill belt for 10 seconds. Immediately after exercise, blood lactate and glucose were measured in a drop of tail blood with portable lactate meter (Lactate Plus, Nova Biomedical) and glucometer (One-Touch Ultra, Lifescan). Lactate and glucose measurements were repeated at 15, 30, and 60 min after exercise cessation. Five days after cessation of exercise, mice were sacrificed for tissue collection.

### Surgical procedures

Catheters were implanted in the jugular vein and carotid artery for stable isotope infusion and sampling protocols as previously described [15,19,20]. The free ends of the catheters were tunneled under the skin to the back of the neck and the exteriorized ends of the vascular catheters were flushed with 200 U/ml of heparinized saline and sealed with stainless-steel plugs. Following the surgical procedures, mice were individually housed and provided ~9–10 days of post-operative recovery prior to stable isotope infusions studies during rest and acute exercise.

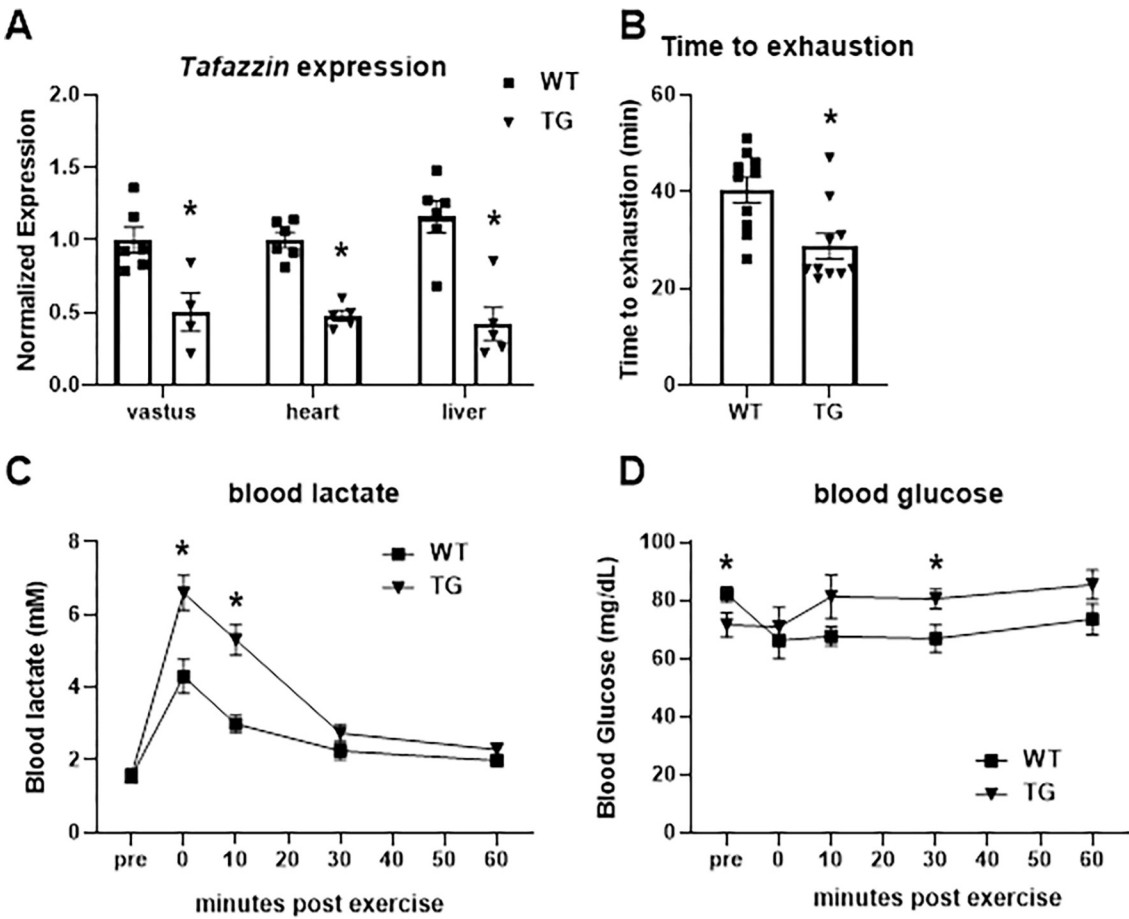

**Fig 1. Exercise tolerance with Tafazzin knockdown. A,** Expression of Tafazzin in vastus, heart, or liver of WT and TG mice after undergoing exercise studies shown in this figure. **B,** Treadmill exercise time to exhaustion for WT and TG mice. **C and D,** a time course of tail blood lactate **(C)** and glucose **(D)** concentrations after exercise. Data are mean ± S.E. *p < 0.05 versus WT at specified time point.

### Stable isotope infusions

The weights of all mice were within 10% of pre-surgery body weight prior to the stable isotope infusions. Food and water were withdrawn within 1 hr of start of light cycle (6–8 am). Two hours into the fast, mice were placed in an enclosed single lane treadmill (Columbus Instruments, Columbus, OH) and the exteriorized catheters were connected to infusion syringes (Fig 2). Three hours into the fast, an 80 μl arterial blood sample was obtained to determine natural isotopic enrichment of plasma glucose. Immediately following this sample, stable isotope infusions were initiated as previously performed [15,20] (Fig 2). Briefly, a $^2H_2O$ (99.9%)-saline bolus containing [6,6-$^2H_2$]glucose (99%) was administered over 25 minutes to both enrich body water and deliver a [6,6-$^2H_2$]glucose prime (440 μmol/kg). This was immediately followed by a continuous infusion of [6,6-$^2H_2$]glucose (4.4 μmol/kg/min). A primed (1.1 mmol/kg), continuous (0.055 mmol/kg/min) intravenous infusion of [U-$^{13}C$]propionate was started two hours after the $^2H_2O$ bolus and [6,6-$^2H_2$]glucose prime. Four 100–150 μl arterial blood samples were obtained (90–120 min following the [$^{13}C_3$] propionate bolus) to determine arterial blood glucose concentration as well as plasma glucose enrichment used in $^2H/^{13}C$ metabolic flux analyses protocols to quantify hepatic glucose and associated nutrient fluxes. The

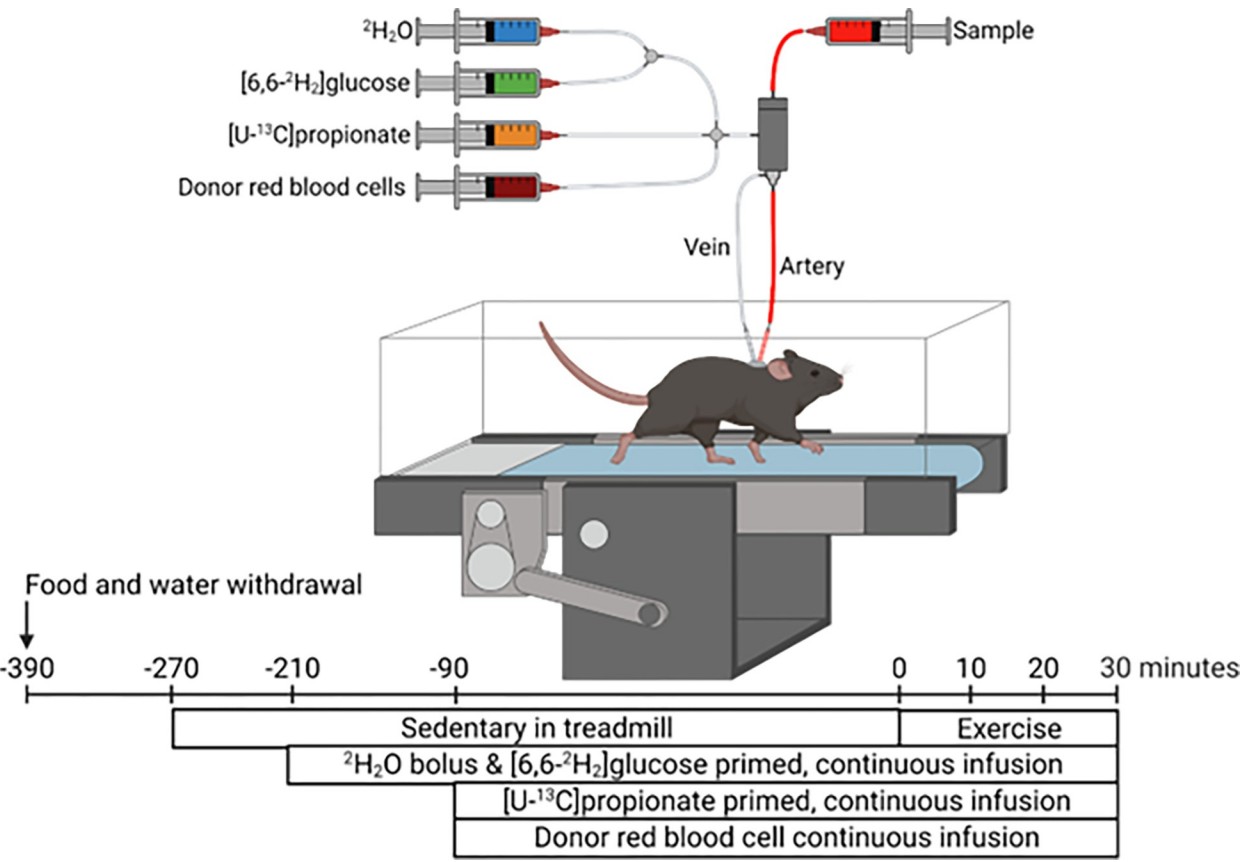

**Fig 2. Study design and data collection.** Stable isotope infusions at rest and during acute treadmill running bout were performed in mice ~9–10 days following carotid arterial and jugular catheter implantation surgeries. At 210 minutes prior to the treadmill running bout (3 hours of fasting), a $^2H_2O$ bolus was administered into the venous circulation to enrich total body water at 4.5%. A [6,6-$^2H_2$]glucose prime was infused followed by a continuous infusion was initiated with the $^2H_2O$ bolus. Ninety minutes before the onset of exercise (5 hours of fasting), a primed, continuous infusion of [U-$^{13}$C]propionate was started. Donor red blood cells were administered to prevent a decline in hematocrit. Arterial samples were obtained prior to stable isotope infusion as well as during 30-minute exercise bout for $^2$H/$^{13}$C metabolic flux analysis.

sample taken at 90 minutes following the [U-$^{13}$C]propionate bolus (Time = 0 min) was obtained while mice were at rest on a stationary treadmill. Samples taken 100–120 minutes following the [U-$^{13}$C]propionate bolus (Time = 10–30 min) were obtained while mice were performing a 30-min acute treadmill running bout at 12 m/min. Donor red blood cells were given by constant rate infusion for the duration of the study to ensure hematocrit did not fall more than 10%. Immediately following the exercise bout, mice were sacrificed via pentobarbital and tissue collected was flash frozen.

## Glucose derivatizations and GC-MS

Forty (40) μl of plasma from the -210-, 0-, 10-, 20-, and 30-minute time points was processed to obtain di-*O*-isopropylidene propionate, aldonitrile pentapropionate, and methyloxime pentapropionate derivatives of glucose as previously described [20]. Briefly, GC-MS injection volumes of 1 μl with purge flow times between 20 and 120 seconds was used. A custom MATLAB function was used to integrate derivative peaks in order to obtain mass isotopomer distributions (MIDs) for six glucose fragment ions. The following fragment ion ranges were used for determining uncorrected MIDs: aldonitrile, m/z 173–178, 259–266, 284–291, and 370–379; methyloxime, m/z 145–149; di-O-isopropylidene, m/z 301–314.

## $^2$H/$^{13}$C metabolic flux analysis

The *$^2$H/$^{13}$C Metabolic Flux Analysis* methodology has been previously described [20]. To summarize, a reaction network was generated using Isotopomer Network Compartmental Analysis (INCA) software. The reaction network defined carbon and hydrogen transitions for hepatic glucose producing and associated oxidative metabolism reactions. Flux through each network reaction was determined relative to citrate synthase flux ($V_{CS}$) by minimizing the sum of squared residuals between simulated and experimentally determined MIDs of the six fragment ions. Flux estimates were repeated 50 times from random initial values, goodness of fit was evaluated by a chi-square test ($p = 0.05$), and confidence intervals of 95% were determined. Mouse body weights and the [6,6-$^2$H$_2$]glucose infusion rate were used to quantify absolute flux values.

## Liver and skeletal muscle glycogen assays

Tissue glycogen was quantified as previously described [21] with minor modifications. Frozen liver and skeletal muscle samples (30–90 mg) were hydrolyzed in 0.3 ml of 30% KOH solution in a boiling water bath for 30 min. Samples were vortexed at 10 and 20 min during the incubation to facilitate digestion. Samples were allowed to cool to room temperature before adding 0.1 ml of 1 M Na2SO4 and 0.8 ml of 100% EtOH. Next, samples were boiled for 5 min then centrifuged at 10,000 X g for 5 min. Liquid was aspirated and the remaining glycogen pellet was dissolved in 0.2 ml of water, followed by two additional ethanol washes. The final glycogen pellet was dried in a speed vacuum and dissolved in 0.2 ml of 0.3 mg/ml amyloglucosidase in 0.2 M sodium acetate buffer (pH 4.8) and incubated for 3 h at 40˚C. The reaction mixture was diluted two- to five-fold with water. Glucose concentration was determined with a glucose assay kit (Sigma-Aldrich; cat# GAGO20).

## Gene expression analyses

Total RNA from cells or tissues was extracted using the RNeasy Lipid Tissue Mini Kit (Qiagen). Real-time quantification of mRNA levels was performed using the PowerSYBR Green PCR master mix (Applied Biosystems) per the manufacturer's instructions. Normalized *Ct* values were subjected to statistical analysis, and fold difference was calculated by the Δ∆*Ct* method. Primer sequences were as follows: *Tafazzin*–fwd: CCC TCC ATG TGA AGT GGC CAT TCC rev: TGG TGG TTG GAG ACG GTG ATA AGG, *Pygl*–fwd: CCT ATG GCT ACG GCA TTC GT rev: TCT CCC AAG GGT TTC CAT GC, *G6pc*–fwd: CAA TCT CCT CTG GGT GGC AGT rev: CAC CAA TAC GGG CGT TGT CC, *Pck1*—fwd: TGG ATG TCG GAA GAG GAC TT rev: TGC AGG CAC TTG ATG AAC TC, Rplp0 (36B4)–fwd: GCA GAC AAC GTG GGC TCC AAG CAG AT rev: GGT CCT CCT TGG TGA ACA CGA AGC CC.

## Glycogen phosphorylase activity assays

Glycogen phosphorylase activity was measured by using frozen liver tissue and the Glycogen Phosphorylase Colorimetric Assay Kit from BioVision (Catalog # K179-100) according the manufacturer's instructions.

## Statistics

Blood glucose concentrations and nutrient fluxes were compared using a repeated measure ANOVA, a Geisser-Greenhouse correction, and Šídák's posthoc test for pairwise comparisons. Other analyses not involving repeated measures were analyzed using a Student's T-test. All data are reported as means ± SEM.

## Results

### Exercise tolerance and plasma metabolite concentrations

As shown in Fig 1A, compared to littermate control mice (also fed doxycycline chow), the expression of Tafazzin was diminished in Tafazzin shRNA transgenic (TG) mice approximately 50% after 6 months of inducing shRNA expression with doxycycline.

Prior work has suggested that transgenic mice with Tafazzin knockdown exhibit impaired exercise performance [22]. To confirm this in our hands, WT (n = 10) and Tafazzin shRNA transgenic (n = 10) underwent a graded exercise performance test on a motorized treadmill after 6 months of transgene induction. Compared to WT littermates, TG mice reached exhaustion sooner in this exercise paradigm (Fig 1B). At the end of the exercise bout, TG mice exhibited higher plasma lactate concentrations compared to WT mice (Fig 1C). Circulating lactate remained elevated 15 min post-exercise, but had returned to WT levels by 30 min (Fig 1C). Plasma glucose concentrations tended to be higher in TG post-exercise and glucose was significantly increased compared to WT mice at 30 min following the completion of exercise (Fig 1D).

### Biometric characteristics and glucose and associated nutrient flux estimates

To better characterize metabolism in these mice during exercise, WT mice (n = 10) and TG (n = 12) mice underwent surgery to implant jugular and carotid catheters and then were studied during treadmill exercise at ~53 weeks of age. However, only 7 WT and 6 TG mice were able to complete the 30-min exercise protocol. The results presented herein include only the data from mice completing the 30-min exercise bout. TG mice exhibited reduced body weight (TG: 27.9 ± 0.6 vs. WT: 33.9 ± 1.3g, p<0.002), which could have contributed to lower work performed during the 30-minute treadmill run (TG: 98.4 ± 2.1 vs. WT: 119.7 ± 12.1J, p<0.002); however due to potential heterogeneity in the model, it is possible that more severe metabolic impairments were present in those not completing the exercise period.

Arterial glucose levels were higher in TG mice compared to WT mice at 20 and 30 minutes of treadmill running (Fig 3B). Consistent with the higher blood glucose was increased endogenous glucose production ($V_{EndoRa}$; Fig 3A and 3C) owing to elevated glycogenolysis ($V_{PYGL}$; Fig 3D) in TG mice compared to WT mice during exercise. In contrast, total gluconeogenesis ($V_{Aldo}$; Fig 3E), gluconeogenesis from glycerol ($V_{GK}$; Fig 3F), and gluconeogenesis from phosphoenolpyruvate ($V_{Enol}$; Fig 3G) were comparable between genotypes at rest and during exercise. Total cataplerosis ($V_{PCK}$; Fig 3H) and anaplerotic fluxes ($V_{LDH}$, $V_{PC}$, and $V_{PCC}$; Fig 3I–3K) were also similar between TG and WT mice at rest and during exercise. However, $V_{LDH}$ trended higher in TG mice during exercise, which is in agreement with higher lactate levels at the conclusion of exercise (Fig 1B). TCA cycle fluxes ($V_{CS}$ and $V_{SDH}$) also trended higher in TG mice during the treadmill run (Fig 3L and 3M). Pyruvate cycling ($V_{PK+ME}$) not different between genotypes at rest and during exercise (Fig 3N). Area under the curve (AUC) for the VLDH, VPC, and VPCC fluxes were also performed. For all fluxes, the AUC from rest to the end of the 30-min exercise bout was similar between genotypes. However, there was a trend towards a higher AUC in TG for VLDH (p = 0.076) (all S1 Fig).

### Hepatic and skeletal muscle glycogen content

Liver and quadriceps glycogen content was assessed in WT and TG mice by using tissue collected immediately after cessation of the exercise experiment described in Fig 3. Interestingly,

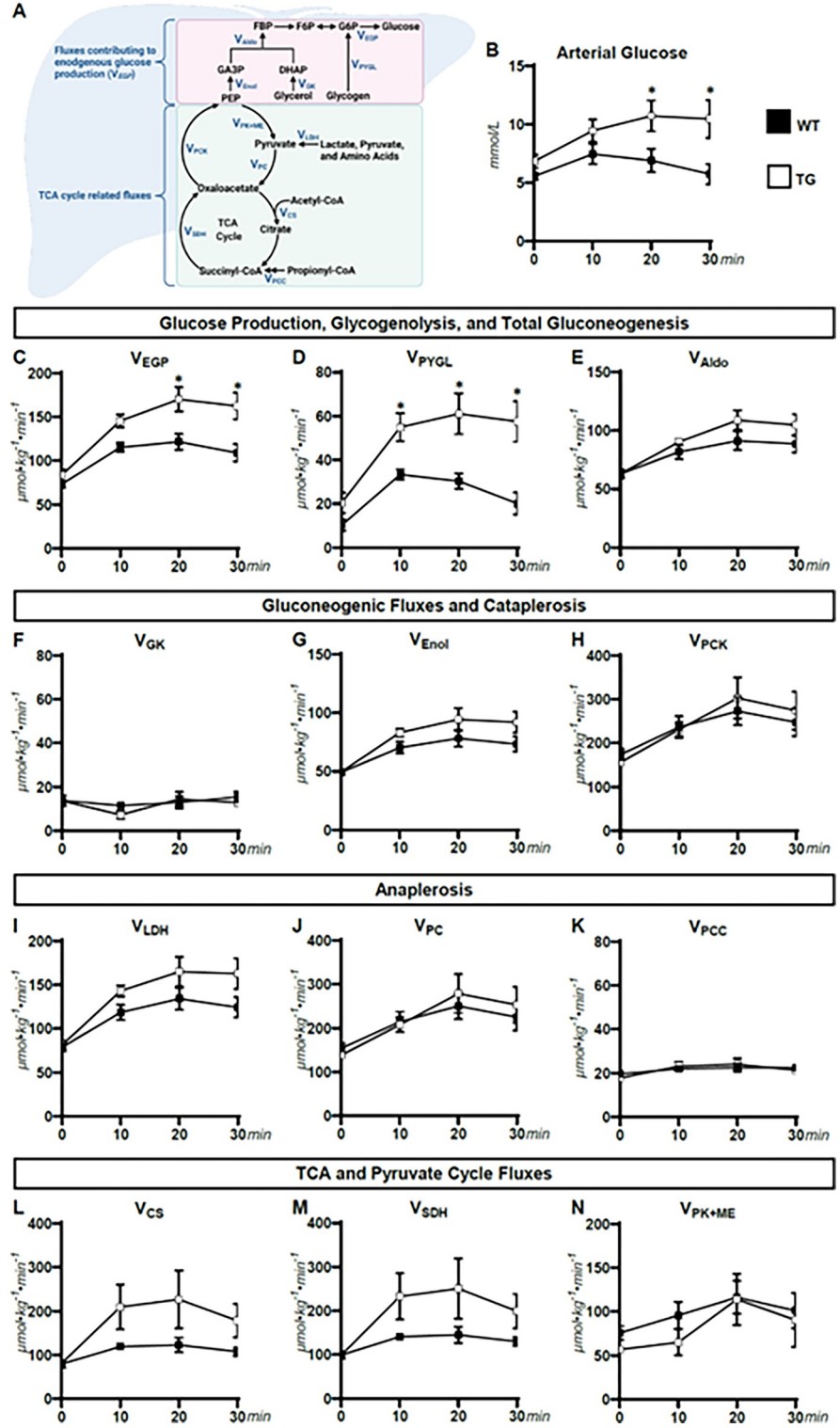

**Fig 3. Nutrient fluxes at rest and during exercise.** **(A)** schematic representation of select glucose producing and tricarboxylic acid cycle fluxes quantified by $^2$H/$^{13}$C metabolic flux analysis. **(B)** A time course of blood glucose concentration (mmol/L) in WT and TG mice prior to (0-minute timepoint) and during a 30-minute treadmill run (10-30-minute timepoints). Model-estimated, nutrient fluxes (μmol·kg$^{-1}$·min$^{-1}$) in WT and TG mice prior to and during a 30-minute of treadmill run for **(C)** endogenous glucose production ($V_{EGP}$), **(D)** glycogenolysis ($V_{PYGL}$), **(E)** total gluconeogenesis ($V_{Aldo}$), **(F)** gluconeogenesis from glycerol ($V_{GK}$), **(G)** gluconeogenesis from phosphoenolpyruvate

($V_{Enol}$), **(H)** total cataplerosis ($V_{PCK}$), **(I)** flux from unlabeled, non-phosphoenolpyruvate, anaplerotic sources to pyruvate ($V_{LDH}$), **(J)** anaplerosis from pyruvate ($V_{PC}$), **(K)** anaplerosis from propionyl-CoA ($V_{PCC}$), **(L)** flux from oxaloacetate and acetyl-CoA to citrate ($V_{CS}$), **(M)** flux from succinyl-CoA to oxaloacetate ($V_{SDH}$), and **(N)** pyruvate cycling ($V_{PK+ME}$) (n = 6–7 per genotype). Data are mean ± SEM. *p<0.05 vs. WT at specified time point by two-way repeated measures ANOVA followed by Šidák's post-hoc tests.

although skeletal muscle glycogen content did not statistically differ by mouse genotype, hepatic glycogen content was significantly higher in TG mice compared to littermate WT mice after exercise (Fig 4A and 4B). We detected no statistical differences in the expression of genes encoding enzymes involved in glycogen phosphorylation (*Pygl*), glucose release (*G6pc*), or gluconeogenesis (*Pck1*) (Fig 4C). Finally, there was no significant difference in hepatic glycogen phosphorylase activity (Fig 4D).

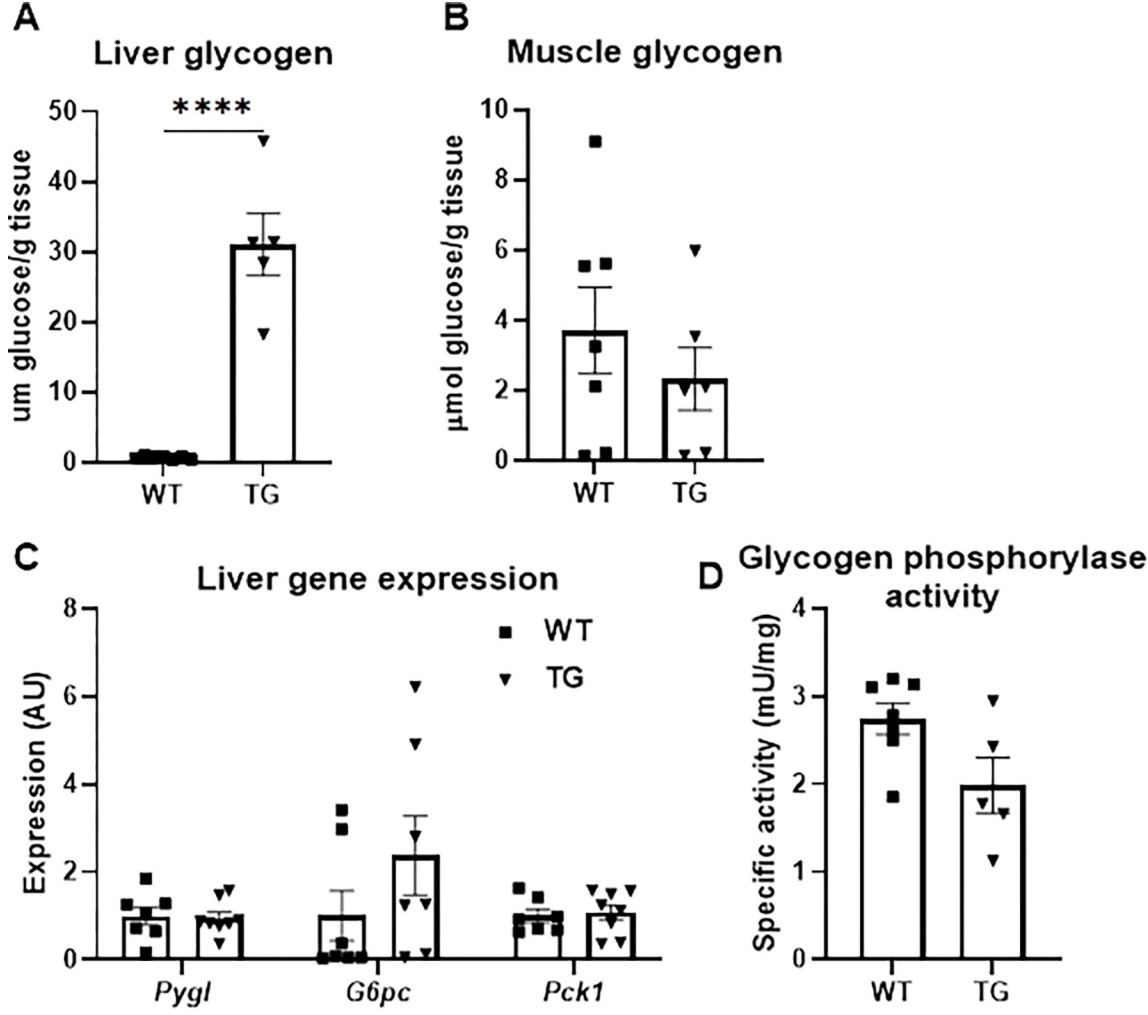

**Fig 4. Hepatic and skeletal muscle glycogen content. A-B,** Glycogen content of liver **(A)** and quadriceps **(B)** tissue collected from WT and TG mice immediately following the 30 minute exercise challenge shown in Fig 3. ****p<0.001 vs. WT **C,** Expression of indicated genes in liver tissue collected from WT and TG mice immediately following the 30 minute exercise challenge shown in Fig 3. **D,** Glycogen phosphorylase activity in liver tissue collected from WT and TG mice immediately following the 30 minute exercise challenge shown in Fig 3.

## Discussion

The purpose of this study was to quantify gluconeogenesis and glycogenolysis in Tafazzin knockdown mice at rest and during acute exercise using stable isotope-based protocols. The primary finding of the study was that, compared to WT mice, Tafazzin knockdown mice exhibited increased endogenous glucose production and increased flux from glycogenolysis during exercise. These findings suggest that Tafazzin knockdown results in increased liver glycogenolysis during exercise and that this metabolic substrate may be preferentially used to support increased anaerobic metabolism during exercise in individuals with BTHS.

Our previous studies have shown that skeletal muscle glucose metabolism is increased in children, adolescents, and adults with BTHS during rest, moderate-intensity exercise, and post-exercise recovery and that myocardial glucose metabolism is higher and associated with lower left ventricular function in adults with BTHS [9,23]. However, these prior studies did not determine which process(es) contributed to or supported these alterations in extrahepatic glucose metabolism during these conditions. The present study extends these findings by showing that higher blood glucose concentration during moderate-intensity exercise in BTHS is mediated at least in part by increased endogenous glucose production via enhanced liver glycogenolysis. In healthy individuals during exercise, the glucoregulatory system maintains blood glucose within a narrow range [24] and glucose release from the liver is increased to maintain blood glucose concentrations despite increased muscle glucose disposal during exercise [25–27]. However, impaired exercise performance in BTHS suggests that increased glycolytic metabolism is insufficient to compensate for impaired mitochondrial function [9,23]. Indeed, fatty acid metabolism during exercise in humans with BTHS is severely blunted [9] and defects in fatty acid oxidation have been demonstrated in the Tafazzin knock down mouse model [18,28] and in cardiac myocytes from individuals with BTHS [29]. Although fatty acid oxidation was not directly measured in the current study, interestingly TCA flux trended higher in the TG group. Although not completely clear, TCA flux in BTHS might remain intact but oxidation is impaired due to respiratory enzyme complex deficiency [28,29].

We hypothesized that both liver glycogenolysis and gluconeogenesis would be upregulated during moderate-intensity exercise in the Tafazzin knockdown compared to WT mice. However, we found that glycogenolysis but not gluconeogenesis was increased. Since gluconeogenesis requires high rates of mitochondrial of mitochondrial flux and ATP production, one might predict that hepatic mitochondrial dysfunction could impair the ability to convert pyruvate or amino acids to new glucose. However, hepatic mitochondrial cataplerotic and anaplerotic fluxes were normal and even slightly increased in the TG mice compared to WT controls suggesting that this does not limit gluconeogenesis. While there is prior evidence of mitochondrial dysfunction in skeletal and cardiac muscle, the present data suggest that hepatic mitochondrial function is not affected in Tafazzin mice. Another recent publication indicated that hepatic cardiolipin content, electron transport chain supercomplex formation, and fatty acid oxidation were not affected in this mouse model [30]. To our knowledge, whether BTHS patients also exhibit preserved hepatic cardiolipin content and mitochondrial metabolism is not known. Compared to heart and skeletal muscle, Tafazzin is expressed at much lower levels in liver of mice [30]. It is possible that the liver has a limited requirement for Tafazzin activity in the regulation of hepatic mitochondrial metabolism.

Interestingly, liver glycogen stores were higher in TG mice compared to WT mice after exercise despite increased rates of glucose production from glycogen throughout the exercise challenge. Indeed, liver glycogen in the WT mice was almost completely depleted after exercise and consistent with this, rates of glycogenolysis tended to decrease as the duration of exercise increased. On the other hand, rates of glycogenolysis in TG mice remained elevated even at 30

minutes and measurable glycogen was still present after 30 minutes of exercise. The likely explanation for this is that TG mice had higher glycogen stores at the beginning of the exercise bout when the mice had been fasting for 6.5 h. However, this was not assessed in our studies.

The use of hepatic glycogen as the primary substrate to meet prolonged energy demands may explain some of the clinical manifestations of BTHS. Studies in individuals with non-BTHS heart failure suggest that decreasing fatty acid metabolism is associated with worsening left ventricular function and decreased myocardial efficiency [31]. We have found a relationship between increased myocardial glucose utilization and worse left ventricular function also in BTHS [23]. Given the impairments in mitochondrial metabolism in BTHS patients, it is likely that cardiac muscle depends more highly on anaerobic glycolysis, instead of mitochondrial pyruvate/lactate or fatty oxidation, to produce ATP. High rates of glycolysis may initially be an adaptive response to cardiac pressure overload; however, in the failing heart, it may be accompanied by reduced mitochondrial oxidation of glucose (as pyruvate), leading to an uncoupling between glucose uptake and oxidation [32]. This imbalance has been implicated in the pathological remodeling of the heart in many forms of acquired cardiomyopathies [33]. Although the present study did not measure metabolic fluxes for cardiac and skeletal muscle glycolysis and glucose oxidation, our data combined with previous studies in humans showing elevated plasma lactate levels [9,10,23], suggest that a disconnect between anaerobic and aerobic metabolism may be occurring in skeletal and cardiac muscle tissue. Indeed, prior work has demonstrated that glucose oxidation rates in heart or skeletal muscle are reduced in the Tafazzin knockdown mouse model [34].

Our data, combined with previous studies, have clinical implications in those with BTHS. These data support the notion that individuals with BTHS compensate with enhanced glucose metabolic processes for deficits in fatty acid metabolism in elevated energy-requiring states and this appears to be related to cardiomyopathy. As chronic, elevated glucose metabolism is associated with heart failure [35], future metabolic therapies targeting intermediary metabolism, including ketone metabolism [36,37], could potentially alleviate cardiomyopathy and improve exercise and activity tolerance in those affected with BTHS. Supplemental amino acid cocktails have been anecdotally provided to BTHS patients with unclear effects and currently there is an ongoing study examining the effects of medium-chain fatty acids on cardiac function and exercise tolerance in the Tafazzin knockout mouse model (personal communication, Barth Syndrome Foundation).

A limitation of this study is the small sample size of mice able to complete the 30-minute bout of exercise. Severe exercise intolerance is well-known in humans with BTHS [7]. Since the Tafazzin-KD model has some residual Tafazzin protein [17] and the effect of residual Tafazzin protein on exercise-induced hepatic glycogenolysis is not known, future studies should also include using the recently-developed floxed Taffazin knockout mouse model which could provide information regarding the effects of Taffazin deficiency in a tissue specific manner [38,39]. Also, plasma leptin has been shown to be lower in Tafazzin-KD mice [10] and since leptin has been shown to have an inhibitory effect on hepatic glycogenolysis and gluconeogenesis [40], it is possible that lower circulating leptin influenced higher hepatic glycogenolysis in our model, however; we did not measure plasma leptin in our study. Additionally, future research is warranted to measure the amount of liver, skeletal muscle and cardiac muscle glycogen before exercise and to examine the specific enzymatic activity of glycogen phosphorylase and phosphoglucomutase, which are involved in the conversion of glycogen to glucose-6-phosphate. It will also be of interest to further characterize effects of Tafazzin deficiency on muscle glucose and lactate metabolism.

In conclusion, we found that enhanced endogenous glucose production via increased liver glycogenolysis contributes to elevated glucose metabolism during moderate intensity exercise

in Tafazzin knockdown mice. Despite this potentially adaptive response to limited mitochondrial capacity, Tafazzin knockdown mice exhibited reduced exercise capacity in a treadmill running exercise paradigm. These findings contribute to our understanding of substrate metabolism in BTHS. Further research into the role of other physiologic processes (i.e. muscle glycogenolysis) in altered glucose metabolism and future studies investigating nutritional and/ or metabolic strategies to improve cardiac function, exercise tolerance and subjective fatigue in BTHS are warranted.

## Supporting information

**S1 Fig. Area under the curve for anaplerotic and TCA cycle fluxes.**
(PPTX)

**S1 File. Raw data for metabolic analyses.**
(XLS)

## Author Contributions

**Conceptualization:** George G. Schweitzer, Curtis C. Hughey, Brian N. Finck, William Todd Cade.

**Data curation:** George G. Schweitzer, Curtis C. Hughey.

**Formal analysis:** George G. Schweitzer, Grace L. Ditzenberger, Curtis C. Hughey, Brian N. Finck, William Todd Cade.

**Funding acquisition:** George G. Schweitzer, Brian N. Finck, William Todd Cade.

**Investigation:** George G. Schweitzer, Curtis C. Hughey, Brian N. Finck, Michael R. Martino.

**Methodology:** George G. Schweitzer, Curtis C. Hughey, Brian N. Finck, Michael R. Martino, William Todd Cade.

**Project administration:** George G. Schweitzer.

**Resources:** Brian N. Finck, William Todd Cade.

**Supervision:** Brian N. Finck, William Todd Cade.

**Writing – original draft:** George G. Schweitzer, Grace L. Ditzenberger, Curtis C. Hughey, Brian N. Finck, William Todd Cade.

**Writing – review & editing:** George G. Schweitzer, Grace L. Ditzenberger, Curtis C. Hughey, Brian N. Finck, Michael R. Martino, Christina A. Pacak, Barry J. Byrne, William Todd Cade.

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
