## [Decision Letter · Decision Letter 0]

10 May 2023

PONE-D-23-09366Elevated Liver Glycogenolysis Mediates Higher Blood Glucose During Acute Exercise in Barth SyndromePLOS ONE

Dear Dr. Cade,

Thank you for submitting your manuscript to PLOS ONE. After careful consideration, we feel that it has merit but does not fully meet PLOS ONE’s publication criteria as it currently stands. Therefore, we invite you to submit a revised version of the manuscript that addresses the points raised during the review process. Additional comments from the academic editor:

1. Please, review and follow the PLOS ONE guidelines for financial disclosure, including whether the funders played any role in the study design, data collection and analysis, decision to publish, or preparation of the manuscript?

2. In the data availability statements, the authors states that "Data are available through Washington University. Please contact Brian Finck, PhD @ bfinck@wustl.edu.". However, authors should provide justifications for not providing the data undelying the study as supplementary file or data deposition in publicly available resources. Please, review and follow PLOS ONE guidelines on data avialbility.  

We look forward to receiving your revised manuscript.

Kind regards,

Elsayed Abdelkreem, MD, PhD

Academic Editor

PLOS ONE

Journal Requirements:

"This study was supported by the Barth Syndrome Foundation, the Vanderbilt University Mouse Metabolic Phenotyping Center for core services (NIDDK, National Institutes of Health (NIH), DK059637), and NIH grants U01 DK094416, P30 DK056341 (Washington University Nutrition Obesity Research Center), P30 DK092950 (Washington University Center for Diabetes Translation Research), P30 DK20579 (Washington University Diabetes Research Center) UL1 TR002345 (Washington University Clinical and Translational Science Award), and R01 DK117657 (Finck)."

"The authors declared no conflict of interest."

6. PLOS requires an ORCID iD for the corresponding author in Editorial Manager on papers submitted after December 6th, 2016. Please ensure that you have an ORCID iD and that it is validated in Editorial Manager. To do this, go to ‘Update my Information’ (in the upper left-hand corner of the main menu), and click on the Fetch/Validate link next to the ORCID field. This will take you to the ORCID site and allow you to create a new iD or authenticate a pre-existing iD in Editorial Manager. Please see the following video for instructions on linking an ORCID iD to your Editorial Manager account: https://www.youtube.com/watch?v=_xcclfuvtxQ.

Reviewers' comments:

Reviewer's Responses to Questions

**Comments to the Author**

1. Is the manuscript technically sound, and do the data support the conclusions?

Reviewer #1: Yes

Reviewer #2: Yes

Reviewer #3: Yes

Reviewer #4: Yes

Reviewer #5: Yes

Reviewer #6: Yes

2. Has the statistical analysis been performed appropriately and rigorously? 

Reviewer #1: Yes

Reviewer #2: Yes

Reviewer #3: Yes

Reviewer #4: Yes

Reviewer #5: Yes

Reviewer #6: Yes

3. Have the authors made all data underlying the findings in their manuscript fully available?

Reviewer #1: Yes

Reviewer #2: Yes

Reviewer #3: Yes

Reviewer #4: Yes

Reviewer #5: Yes

Reviewer #6: Yes

4. Is the manuscript presented in an intelligible fashion and written in standard English?

Reviewer #1: Yes

Reviewer #2: Yes

Reviewer #3: Yes

Reviewer #4: Yes

Reviewer #5: Yes

Reviewer #6: Yes

5. Review Comments to the Author

Reviewer #1: Barth Syndrome s cause by a mutation in the Tafazzin gene. It is known that Barth Syndrome patients exhibit defects in muscle mitochondrial oxidative metabolism which results in enhanced reliance on anaerobic metabolism during exercise. The impact of Tafazzin deficiency on hepatic glucose production and the pathways contributing to hepatic glucose production during exercise was unknown. Thus, the objective of this study was to examine in vivo liver gluconeogenesis and glycogenolysis in Tafazzin knockdown mice at rest and during acute exercise. They performed stable substrate isotope infusions at rest and during a 30-minute treadmill running exercise and quantified hepatic glucose production and associated nutrient fluxes and compared this to sedentary conditions. They show that Tafazzin knockdown mice exhibit elevated anaerobic metabolism during rest and exercise and higher hepatic glycogenolysis. Interestingly, the authors found that glycogenolysis but not gluconeogenesis was increased which is not what one would expect and is an intriguing finding. Overall an excellent well written and presented manuscript with appropriate controls and statistical analysis (n=10 per group). The conclusions appear to be justified by the data.

Major comments:

1. Since Tafazzin knockdown mice exhibit residual Tafazzin protein, as opposed to Barth Syndrome patients which exhibit no functional Tafazzin protein, the authors need to address this limitation and implications of this in the Discussion of their results. Additionally, do the authors have information on the cardiolipin/lysocardiolipin levels in these animals as the altered ratio is a hallmark of Barth Syndrome.

2. Do the authors have any information on the body weights of the animals before an after treatment? Tafazzin knockdown mice also have lower hepatic basal respiration, are hypermetabolic and generally have less weight gain than wild type mice (Ref 29) and this might affect exercise and hepatic metabolic parameters. The authors could discuss these implications.

3. Tafazzin knockdown mice also exhibit reduced plasma glucose as well as leptin levels (Ref 29) and leptin increases inhibitory effects of insulin on gluconeogenesis and glycogenolysis in liver (Morton et al 2011). Could the authors comment on the potential impact of this on the exercise-induced elevated hepatic glycogenolysis?

Reviewer #2: Manuscript #: PONE-D-23-09366

Title: Elevated Liver Glycogenolysis Mediates Higher Blood Glucose During Acute Exercise in Barth Syndrome

BS behaves like any mitochondrial disorder. They have to rely on anaerobic metabolism, which requires gluconeogenesis.

Therefore, the results are not new and knonw from other models of mitochondrial disease

4/23

Reviewer #3: Study uses well-designed methods to assess anaerobic metabolism during exercise in Tafazzin knockdown animals. Findings provide important insight supporting the hypothesis that increased liver glycogenolysis occurs in Taz knockdown animals to compensate for reduced fatty acid oxidation in muscle. Concerns are minor in nature.

1. I agree with authors that it is not surprising that several Taz knockdown animals could not complete the 30 minute exercise challenge. It is perhaps more surprising that only 70% of the controls animals completed the challenge. Would be worthwhile for authors to comment on previous results with similar challenges in these control mice. Is this a normal rate of failure, or something unusual? Is it possible that running with the catheter impairs the exercise and contributes to the failure rate? Would this affect interpretation?

2. Authors should comment on the efficiency of shRNA knockdown here, as variation in knockdown efficiency could contribute to phenotypic variation? Was knockdown confirmed by qRT-PCR?

3. Is it possible that other acyl-transferases are active in mouse liver that can compensate for reduced taz? Lower expression level of tafazzin in control liver would be consistent with that. Will be important in future to determine whether human taz expression is also lower in human liver than human muscle and perhaps liver-specific knockouts using the floxed mouse would be informative in elucidating the role of the liver in exercise compensation. These are beyond the scope of the current study however.

There are a few scattered typos that look like cut and paste errors:

Typo in abstract line 57-58

Typo in line 279

Piece missing in sentence on lines 220-221

Typo in line 324

Reviewer #4: The authors present the first investigation of in vivo metabolic flux in the Taz shRNA mouse model of BTHS, which corroborates and extends previous findings in human BTHS patients and static measurements of metabolic rewiring in the same mouse model. The writing, experimental approach and data are of high quality, and the conclusions are well supported by the results and discussed in the context of the most relevant literature in the field. The insights gained from these studies are important and highly relevant for the study of BTHS, and further validate the use of this animal model for studies of BTHS metabolism. Below are a few comments and suggestions for the authors consideration.

1) Abstract Line 34: consider add “patients” after BTHS to note that these data were obtained from humans, thereby further emphasizing the authors’ novel investigation of metabolic flux in this mouse model.

2) Line 163, remove “in a” and please clarify exactly how blood was sample while mice were running, or about how much time elapsed between actual running and sampling (if any). Were the catheter lines connected and suspended above the mice while running as indicated in Figure 2?

3) How do the authors interpret the higher V-CS and V-SDH flux in TG mice if mitochondrial OXPHOS is thought to be impaired in these animals? Might there be specific defects in the oxidation of specific substrates (e.g., fatty acids) that limit exercise performance in the TG mice, but overall mitochondrial oxidative capacity is intact or enhanced?

4) Were the exercise test data presented in Figure 1 obtained from the same mice used in the metabolic flux experiments? Also, please note the age of these animals as was done for instrumented mice (~53 weeks).

5) The authors should note why food was removed specifically for 18 hours prior to exercise testing. Some might expect this to deplete liver glycogen at least partially, so this should be discussed, and perhaps addressed further per the comments below.

6) Line 245: it is stated that “hepatic glycogen content was significantly increased in TG mice compared to littermate WT mice after exercise (Figure 4)”. However, these appear to be static measures obtained from mice sacrificed after exercise, so the appropriate term would by “higher” not “increased”. “Increased” implies that hepatic glycogen increased in TG mice after exercise (i.e., relative to pre-exercise), which probably wasn’t the case. Please clarify.

7) Related to the comment above, it would be valuable to also present liver and muscle glycogen content from TG and WT mice before (fed state) and following the same 18 hour fast, but prior to the exercise test. This would enable a rough determination of how tissue glycogen levels change in these tissues in response to fasting and from pre- to post-exercise, both of which may be relevant for interpreting the metabolic flux data during exercise. The author allude to this in the Discussion (lines 296-298), so including these data would add significant strength to the study.

Reviewer #5: Elevated Liver Glycogenolysis Mediates Higher Blood Glucose During Acute Exercise in Barth Syndrome

Summary:

The Cade Laboratory has previously determined that the defects in muscle mitochondrial oxidative metabolism in Barth Syndrome necessitate enhanced reliance on anaerobic glycolysis during exercise. However, the contributions of hepatic glycogenolysis and gluconeogenesis to hepatic glucose production during exercise in Tafazzin deficiency have not been directly studied. The authors used stable isotope infusions at rest and during a forced treadmill exercise protocol to measure flux through these pathways in shRNA-transgenic Tafazzin knockdown mice, and conclude that higher liver glycogen content and higher glycogenolysis is the primary mechanism supporting increased hepatic glucose production and anaerobic glycolysis during exercise in Tafazzin knockdown mice, while gluconeogenesis and other related pathways are largely unchanged.

This work extends previous findings in human populations by this group, and provides important mechanistic insight. The studies are of a high quality, and I only have a few minor suggestions/comments.

Comments:

1. Please give details on animal ethics approval (e.g. protocol numbers, date approved) in the methods, if available.

2. If the mice were fasted for 18 hours before testing, this could affect stress responses, and could cause glycogen depletion in the controls. A short description in the “Exercise Challenge” section of the Materials and Methods indicating why a fast of this duration was used would be helpful for the reader. This section could also cite previous studies that have used a similar protocol, and the make/model of the portable glucometer and lactate meter used in this protocol should be given.

3. The wording in line 165 is awkward, “did not fall greater than” – suggest: “did not fall by more than . . . ?”

4. The Materials and Methods “Liver and Skeletal Muscle Glycogen Assays” section should provide a citation for this protocol.

5. Figures 1B and 1C would be clearer if the x-axis label read “Time post-exercise (min).” Also, calculation of incremental area-under-the-curve might give a fuller indication of the relative differences in overall responses of mice in different groups, and should be considered.

6. The illustration in Figure 2 is very helpful, thank you.

7. It is suggested that in Figure 3, incremental area-under-the-curve quantitation could help to account for individual variation and would provide a measure of the overall response, and may identify differences that are not apparent by 2-way ANOVA at individual time-points. In particular, this may help to unmask differences between groups for measures in figures 3I, 3L, 3M and 3N, which would be interesting.

8. In Line 244, it is not strictly accurate to say, “although skeletal muscle glycogen content was not affected by mouse genotype.” Rather, the data indicate that skeletal muscle glycogen content did not differ in a statistically significant manner between genotypes – but the average in WT is close to 2-fold greater than in TG, so genotype does appear to affect this measure, it just does not reach a threshold for statistical significance with the power available in this experiment.

Reviewer #6: The manuscript by Schweitzer and co-workers presents an interesting link between glucose metabolism and hepatic glycogenolysis in the BTHS model. The rationale for this study is based on previously reported findings that demonstrate adolescent BTHS patients exhibit impaired fatty-acid oxidation and increased glucose metabolism during exercise (Cade et al., 2019). The current study analyzed gluconeogenesis and glycogenolysis during exercise. Using the TAFAZZIN-knockdown mouse model of BTHS, they tested the rate of flux through gluconeogenesis and glycogenolysis during moderate exercise. It was shown that glycogenolysis but not gluconeogenesis was increased in TAFAZZIN knockdown mice during exercise. Interestingly, after exercise, TAFAZZIN-knockdown mice displayed higher liver glycogen stores. These novel findings indicate an important link between hepatic glycogenolysis and global energy metabolism in BTHS. The paper contributes significantly to our understanding of metabolic perturbation in BTHS.

The following concerns should be addressed:

Major concerns:

1. Why were the mice subjected to isotope infusions twice, once at rest and again during a 30-minute running bout?

2. In the biometric characteristics and glucose and associated nutrient flux estimates, why did almost half of the mice not complete the 30 min exercise protocol?

3. Lines 225-226: The authors state that the weight disparity resulted in lower exercise performance. Insufficient data are presesnted to support this conclusion. It could be due to altered cellular metabolism as observed in many BTHS models and patients.

4. Hepatic and skeletal muscle glycogen content: Clearly, TG liver present with increased glycogen phosphorylase (PYGL), suggesting that TG mice present with increased degradation of glycogen. However, in line 245, how can TG mice exhibit greatly increased levels of glycogen in the liver soon after exercise? Measurement of glycogen labeled would be helpful.

5. Lines 280-281: the authors state that “one possibility is that hepatic mitochondrial dysfunction partially impairs the ability to convert pyruvate or amino acids to new glucose”. There is no evidence to support this statement. The evidence described in the paper does not suggest that hepatic mitochondria suffer from disruptions in energy production. How can it be explained that mice that performed exercise for 30 min and struggle to use energy substrates in cardio-skeletal muscles do not present increased hepatic glucogenesis?

6. Can the authors provide data on the concentration of muscle lactate and glucose? Additionally, while Fig 1B demonstrates that circulating lactate returned to WT levels after 30 minutes, it does not provide comparative data on the process of gluconeogenesis for muscle and blood. This comparison would provide valuable insight into the metabolism of glycogen and glucose in both tissues.

7. Have the authors conducted nutrient flux measurements at rest and during exercise using stable glycogen isotope tracer? These measurements would provide detailed information on the utilization and turnover of glycogen during exercise and at rest.

8. Have the authors investigated the expression levels of enzymes responsible for glycogenolysis to glucose, such as glucose phosphorylase, phosphoglucomutase, and glucose-6-phosphatase? This information would provide valuable insight into the underlying molecular mechanisms. Moreover, it would also be valuable to investigate the relative proportion of isotope labeled metabolites during both resting and exercise conditions in conjunction with their corresponding flux rates.

9. Have glucose tolerance tests or insulin resistance tests been conducted at rest and during exercise? Such tests would allow for the assessment of insulin and glucose uptake efficiency, as well as the effect of exercise on blood glucose levels.

Minor concerns:

1. Line 68-69, the use of “mature” and “immature” cardiolipin is not meaningful. Consider remodeled and unremodeled cardiolipin.

2. Figure 1a. should be more detailed to better indicate that this is the running regiment for each individual mouse.

3. Figure 1B and 1C: ‘Plasma lactate’ is unclear. Does the y axis indicate post-exercise plasma lactate?

4. Figure 3: The title of each figure should indicate that the glucose concentration was measured was during exercise.

5. Line 146: was total body weight recovered after surgery or just 10% recovered?

6. Line 15: What are MFA protocols?

7. Last sentence of the second paragraph, lines 272-275: References should be cited with regard to “prior findings from our group and others support…”

6. PLOS authors have the option to publish the peer review history of their article (what does this mean?). If published, this will include your full peer review and any attached files.

Reviewer #1: No

Reviewer #2: No

Reviewer #3: No

Reviewer #4: No

Reviewer #5: No

Reviewer #6: No

---

## [Author Response · Author response to Decision Letter 0]

31 Jul 2023

Review Comments to the Author

Reviewer #1: Barth Syndrome s cause by a mutation in the Tafazzin gene. It is known that Barth Syndrome patients exhibit defects in muscle mitochondrial oxidative metabolism which results in enhanced reliance on anaerobic metabolism during exercise. The impact of Tafazzin deficiency on hepatic glucose production and the pathways contributing to hepatic glucose production during exercise was unknown. Thus, the objective of this study was to examine in vivo liver gluconeogenesis and glycogenolysis in Tafazzin knockdown mice at rest and during acute exercise. They performed stable substrate isotope infusions at rest and during a 30-minute treadmill running exercise and quantified hepatic glucose production and associated nutrient fluxes and compared this to sedentary conditions. They show that Tafazzin knockdown mice exhibit elevated anaerobic metabolism during rest and exercise and higher hepatic glycogenolysis. Interestingly, the authors found that glycogenolysis but not gluconeogenesis was increased which is not what one would expect and is an intriguing finding. Overall an excellent well written and presented manuscript with appropriate controls and statistical analysis (n=10 per group). The conclusions appear to be justified by the data.

Major comments:

1. Since Tafazzin knockdown mice exhibit residual Tafazzin protein, as opposed to Barth Syndrome patients which exhibit no functional Tafazzin protein, the authors need to address this limitation and implications of this in the Discussion of their results. Additionally, do the authors have information on the cardiolipin/lysocardiolipin levels in these animals as the altered ratio is a hallmark of Barth Syndrome.

Response: We have now added a statement in the Discussion (lines 369-371) regarding the limitation of the TAZ-KD model having some residual protein compared to humans with BTHS. We did not measure MLCL/CL ratio in our mouse model as the increase in MLCL/CL in both heart and skeletal muscle in this model has been well established (PMID: 21068380, reference #17) and this was not a primary focus of our study.

2. Do the authors have any information on the body weights of the animals before and after treatment? Tafazzin knockdown mice also have lower hepatic basal respiration, are hypermetabolic and generally have less weight gain than wild type mice (Ref 29) and this might affect exercise and hepatic metabolic parameters. The authors could discuss these implications.

Response: TG mice exhibited reduced body weight (TG: 27.9 ± 0.6 vs. WT: 33.9 ± 1.3g, p<0.002). This information is provided on lines 255-257.

3. Tafazzin knockdown mice also exhibit reduced plasma glucose as well as leptin levels (Ref 29) and leptin increases inhibitory effects of insulin on gluconeogenesis and glycogenolysis in liver (Morton et al 2011). Could the authors comment on the potential impact of this on the exercise-induced elevated hepatic glycogenolysis?

Response: 

We did not measure plasma leptin in our study and unfortunately do not have mice, tissue, or plasma available to do so now. However, we have now added a statement regarding the potential effect of lower circulating leptin on exercise-induced hepatic glycogenolysis (Lines 371-374). 

Reviewer #2: Manuscript #: PONE-D-23-09366

Title: Elevated Liver Glycogenolysis Mediates Higher Blood Glucose During Acute Exercise in Barth Syndrome

BS behaves like any mitochondrial disorder. They have to rely on anaerobic metabolism, which requires gluconeogenesis.

Therefore, the results are not new and knonw from other models of mitochondrial disease

Response: We respectfully disagree. Although BTHS shares similar characteristics with other mitochondrial diseases, it is distinct due to several characteristics including primary cardiomyopathy, elevated MLCL/CL ratio, and chronic neutropenia. Due to the extremely rare nature of BTHS, it is important to characterize the metabolic phenotype of this condition in order to develop potential therapies. In addition, our results show that gluconeogenesis was not elevated in Barth syndrome which appears different than other mitochondria associated disorders.

Reviewer #3: Study uses well-designed methods to assess anaerobic metabolism during exercise in Tafazzin knockdown animals. Findings provide important insight supporting the hypothesis that increased liver glycogenolysis occurs in Taz knockdown animals to compensate for reduced fatty acid oxidation in muscle. Concerns are minor in nature.

1. I agree with authors that it is not surprising that several Taz knockdown animals could not complete the 30 minute exercise challenge. It is perhaps more surprising that only 70% of the controls animals completed the challenge. Would be worthwhile for authors to comment on previous results with similar challenges in these control mice. Is this a normal rate of failure, or something unusual? Is it possible that running with the catheter impairs the exercise and contributes to the failure rate? Would this affect interpretation?

Response: The inability of 30% of WT mice (3 of 10) to complete a 30-minute exercise bout at 12m/min is higher than our prior observations. Following the same protocols in a previously published study (PMID: 36351254), all WT mice studied completed the exercise protocol in our hands. These experiments are technically challenging. They include stable isotope infusions, donor red blood cell infusions, and arterial sampling in real time while a mouse is running on a treadmill. Given this, it is possible that the protocol may influence exercise performance. However, there are unique variables in this study that may warrant further consideration. First, the mice were approximately 1 year of age where as our prior studies used mice that were ≤15 weeks of age. It is possible that the decline in exercise performance with age in mice contributed to the outcomes of this study. Second, doxycycline in the diet was used to induce the shRNA that underlies the knockdown of Tafazzin. Prior work has showed that antibiotics reduce voluntary exercise behavior and metabolic adaptations to exercise training (PMID: 35504410 and 37056044). 

2. Authors should comment on the efficiency of shRNA knockdown here, as variation in knockdown efficiency could contribute to phenotypic variation? Was knockdown confirmed by qRT-PCR?

Response: Yes. We measured the expression of the gene encoding Tafazzin in mice after long term induction of shRNA expression. We now show that Tafazzin gene expression was diminished about 50% in heart, skeletal muscle (vastus), and liver in transgenic mice compared to wild-type littermate controls. These data can now be found in Figure 1A.

3. Is it possible that other acyl-transferases are active in mouse liver that can compensate for reduced taz? Lower expression level of tafazzin in control liver would be consistent with that. Will be important in future to determine whether human taz expression is also lower in human liver than human muscle and perhaps liver-specific knockouts using the floxed mouse would be informative in elucidating the role of the liver in exercise compensation. These are beyond the scope of the current study however.

Response: Thank you for this very interesting point. We agree it is possible and potentially important that other acyl-transferases might partially compensate for reduced Tafazzin however we did not measure this in the current study. 

There are a few scattered typos that look like cut and paste errors:

Typo in abstract line 57-58

Response: Corrected.

Typo in line 279

Response: Corrected.

Piece missing in sentence on lines 220-221

Response: Corrected.

Typo in line 324

Response: Corrected.

Reviewer #4: The authors present the first investigation of in vivo metabolic flux in the Taz shRNA mouse model of BTHS, which corroborates and extends previous findings in human BTHS patients and static measurements of metabolic rewiring in the same mouse model. The writing, experimental approach and data are of high quality, and the conclusions are well supported by the results and discussed in the context of the most relevant literature in the field. The insights gained from these studies are important and highly relevant for the study of BTHS, and further validate the use of this animal model for studies of BTHS metabolism. Below are a few comments and suggestions for the authors consideration.

1) Abstract Line 34: consider add “patients” after BTHS to note that these data were obtained from humans, thereby further emphasizing the authors’ novel investigation of metabolic flux in this mouse model.

Response: We have added “in humans with BTHS” to indicate the novel investigation of metabolic flux in the Tafazzin KD model (line 35).

2) Line 163, remove “in a” and please clarify exactly how blood was sample while mice were running, or about how much time elapsed between actual running and sampling (if any). Were the catheter lines connected and suspended above the mice while running as indicated in Figure 2?

Response: The words “in a” were removed and the statement “The free ends of the catheters were tunneled under the skin to the back of the neck” prior to “the exteriorized…..” was added under “Surgical Procedures” (Lines 145-146). Samples were taken during exercise (i.e. no time elapsed) as denoted in the statement “Samples taken 100-120 minutes following the [U-13C]propionate bolus (Time = 10-30 min) were obtained while mice were performing a 30-min acute treadmill running bout at 12 m/min” (lines 164-165).

3) How do the authors interpret the higher V-CS and V-SDH flux in TG mice if mitochondrial OXPHOS is thought to be impaired in these animals? Might there be specific defects in the oxidation of specific substrates (e.g., fatty acids) that limit exercise performance in the TG mice, but overall mitochondrial oxidative capacity is intact or enhanced?

Response: We have added the statement in the Discussion to read “Indeed, fatty acid metabolism during exercise in humans with BTHS is severely blunted[9] and defects in fatty acid oxidation have been demonstrated in the Tafazzin knock down mouse model[18, 27] and in cardiac myocytes from individuals with BTHS[28]. Although fatty acid oxidation was not directly measured in the current study, interestingly TCA flux trended higher in the TG group. Although not completely clear, TCA flux in BTHS might remain intact but oxidation is impaired due to respiratory enzyme complex deficiency[27, 28]” (lines 309-314).

4) Were the exercise test data presented in Figure 1 obtained from the same mice used in the metabolic flux experiments? Also, please note the age of these animals as was done for instrumented mice (~53 weeks).

Response: Thank you for catching this oversight. The data from Figure 1 are from different mice that were induced for 6 months at the time of experimentation. Those mice were sacrificed 5 days after the end of the exercise bout. We have now clarified this in the text (lines 124-125). The reason for the differences in age are due to the necessity of transferring mice to Vanderbilt, time in quarantine, as well as surgical prep and recovery.

5) The authors should note why food was removed specifically for 18 hours prior to exercise testing. Some might expect this to deplete liver glycogen at least partially, so this should be discussed, and perhaps addressed further per the comments below.

Response: The 18 h fast had multiple purposes. Fasting before exercise is very common in exercise physiology testing to remove the effect of prior meals on exercise performance. It is well known that macronutrient consumption enhances exercise performance and we wished to minimize individual variability. Secondly, we wished to test our hypothesis that gluconeogenesis would be different and depleting liver glycogen with an 18-hour fast would place greater reliance on gluconeogenesis to possibly enhance differences between genotypes. We added text to explain this (lines 128-131).

6) Line 245: it is stated that “hepatic glycogen content was significantly increased in TG mice compared to littermate WT mice after exercise (Figure 4)”. However, these appear to be static measures obtained from mice sacrificed after exercise, so the appropriate term would by “higher” not “increased”. “Increased” implies that hepatic glycogen increased in TG mice after exercise (i.e., relative to pre-exercise), which probably wasn’t the case. Please clarify.

Response. We agree; “increased” was changed to “higher”.

7) Related to the comment above, it would be valuable to also present liver and muscle glycogen content from TG and WT mice before (fed state) and following the same 18 hour fast, but prior to the exercise test. This would enable a rough determination of how tissue glycogen levels change in these tissues in response to fasting and from pre- to post-exercise, both of which may be relevant for interpreting the metabolic flux data during exercise. The author alludes to this in the Discussion (lines 296-298), so including these data would add significant strength to the study.

Response: We agree that pre-exercise hepatic glycogen content would add to the interpretation of tissue glycogen changes pre-post exercise however animals were sacrificed following the exercise period and we did not have an additional pre-exercise only group to compare to therefore we do not have these data.

Reviewer #5: Elevated Liver Glycogenolysis Mediates Higher Blood Glucose During Acute Exercise in Barth Syndrome

Summary:

The Cade Laboratory has previously determined that the defects in muscle mitochondrial oxidative metabolism in Barth Syndrome necessitate enhanced reliance on anaerobic glycolysis during exercise. However, the contributions of hepatic glycogenolysis and gluconeogenesis to hepatic glucose production during exercise in Tafazzin deficiency have not been directly studied. The authors used stable isotope infusions at rest and during a forced treadmill exercise protocol to measure flux through these pathways in shRNA-transgenic Tafazzin knockdown mice, and conclude that higher liver glycogen content and higher glycogenolysis is the primary mechanism supporting increased hepatic glucose production and anaerobic glycolysis during exercise in Tafazzin knockdown mice, while gluconeogenesis and other related pathways are largely unchanged.

This work extends previous findings in human populations by this group, and provides important mechanistic insight. The studies are of a high quality, and I only have a few minor suggestions/comments.

Comments:

1. Please give details on animal ethics approval (e.g. protocol numbers, date approved) in the methods, if available.

Response: Studies conducted at Washington University were approved under IACUC protocol number 20170034 (approved March 21, 2017). Studies conducted at Vanderbilt were approved under protocol number M2000050-00. This information has been added to the Methods section (line 111).

2. If the mice were fasted for 18 hours before testing, this could affect stress responses, and could cause glycogen depletion in the controls. A short description in the “Exercise Challenge” section of the Materials and Methods indicating why a fast of this duration was used would be helpful for the reader. This section could also cite previous studies that have used a similar protocol, and the make/model of the portable glucometer and lactate meter used in this protocol should be given.

Response: The 18 h fast had multiple purposes. Fasting before exercise is very common in exercise physiology testing to remove the effect of prior meals on exercise performance. It is well known that macronutrient consumption enhances exercise performance and we wished to minimize individual variability. Secondly, we wished to test our hypothesis that gluconeogenesis would be different and depleting liver glycogen with an 18-hour fast would place greater reliance on gluconeogenesis to possibly enhance differences between genotypes. The make and model of glucometer and lactate meter used has now been added.

3. The wording in line 165 is awkward, “did not fall greater than” – suggest: “did not fall by more than . . . ?”

Response: We have replaced “greater” with “more” to now read “Donor red blood cells were given by constant rate infusion for the duration of the study to ensure hematocrit did not fall more than 10%”.

4. The Materials and Methods “Liver and Skeletal Muscle Glycogen Assays” section should provide a citation for this protocol.

Response: Thank you. We have added a citation for this protocol.

5. Figures 1B and 1C would be clearer if the x-axis label read “Time post-exercise (min).” Also, calculation of incremental area-under-the-curve might give a fuller indication of the relative differences in overall responses of mice in different groups, and should be considered.

Response: The x-axis label has been changes to minutes post-exercise as recommended.

6. The illustration in Figure 2 is very helpful, thank you.

Response: Thank you for the encouraging comment.

7. It is suggested that in Figure 3, incremental area-under-the-curve quantitation could help to account for individual variation and would provide a measure of the overall response, and may identify differences that are not apparent by 2-way ANOVA at individual time-points. In particular, this may help to unmask differences between groups for measures in figures 3I, 3L, 3M and 3N, which would be interesting.

Response: We have assessed the area under the curve (AUC) for the fluxes in figures 3I, 3L, 3M, and 3N. For all fluxes, the AUC from rest to the end of the 30-min exercise bout was similar between genotypes. However, there was a trend towards a higher AUC for VLDH (p=0.076). These data in included in Supplementary Figure 1.

8. In Line 244, it is not strictly accurate to say, “although skeletal muscle glycogen content was not affected by mouse genotype.” Rather, the data indicate that skeletal muscle glycogen content did not differ in a statistically significant manner between genotypes – but the average in WT is close to 2-fold greater than in TG, so genotype does appear to affect this measure, it just does not reach a threshold for statistical significance with the power available in this experiment.

Response: We agree. The statement now reads “Interestingly, although skeletal muscle glycogen content did not statistically differ by mouse genotype, hepatic glycogen content was significantly higher in TG mice compared to littermate WT mice after exercise (Figure 4).”

Reviewer #6: The manuscript by Schweitzer and co-workers presents an interesting link between glucose metabolism and hepatic glycogenolysis in the BTHS model. The rationale for this study is based on previously reported findings that demonstrate adolescent BTHS patients exhibit impaired fatty-acid oxidation and increased glucose metabolism during exercise (Cade et al., 2019). The current study analyzed gluconeogenesis and glycogenolysis during exercise. Using the TAFAZZIN-knockdown mouse model of BTHS, they tested the rate of flux through gluconeogenesis and glycogenolysis during moderate exercise. It was shown that glycogenolysis but not gluconeogenesis was increased in TAFAZZIN knockdown mice during exercise. Interestingly, after exercise, TAFAZZIN-knockdown mice displayed higher liver glycogen stores. These novel findings indicate an important link between hepatic glycogenolysis and global energy metabolism in BTHS. The paper contributes significantly to our understanding of metabolic perturbation in BTHS.

The following concerns should be addressed:

Major concerns:

1. Why were the mice subjected to isotope infusions twice, once at rest and again during a 30-minute running bout?

Response: We apologize if this was not clear in the manuscript. Mice were only subjected to the isotope infusion once: the rest period was prior to the exercise period and the stable isotope infusion was continuous throughout both rest and exercise periods.

2. In the biometric characteristics and glucose and associated nutrient flux estimates, why did almost half of the mice not complete the 30 min exercise protocol?

Response: Exercise intolerance is a hallmark in humans with BTHS and also in the Tafazzin-KD model (PMID: 21068380). See response also to previous reviewer.

3. Lines 225-226: The authors state that the weight disparity resulted in lower exercise performance. Insufficient data are presented to support this conclusion. It could be due to altered cellular metabolism as observed in many BTHS models and patients.

Response: We agree. The statement now reads “The results presented herein include only the data from mice completing the 30-min exercise bout. TG mice exhibited reduced body weight (TG: 27.9 ± 0.6 vs. WT: 33.9 ± 1.3g, p<0.002), which could have contributed to lower work performed during the 30-minute treadmill run (TG: 98.4 ± 2.1 vs. WT: 119.7 ± 12.1J, p<0.002); however due to potential heterogeneity in the model, it is possible that more severe metabolic impairments were present in those not completing the exercise period.” (lines 254-259)

4. Hepatic and skeletal muscle glycogen content: Clearly, TG liver present with increased glycogen phosphorylase (PYGL), suggesting that TG mice present with increased degradation of glycogen. However, in line 245, how can TG mice exhibit greatly increased levels of glycogen in the liver soon after exercise? Measurement of glycogen labeled would be helpful.

Response: Indeed, the large difference in post-exercise liver glycogen between genotypes provokes many questions. We believe that pre-exercise liver glycogen content would add to the interpretation of differences in liver glycogen pre-post exercise. However, mice were sacrificed following the exercise period and we did not include an additional pre-exercise only group in our study design. Without knowing if the pre-exercise glycogen levels are different, the labeling of liver glycogen would be very challenging to interpret because the inner glycogen core (i.e. glycogen present prior to isotope infusion) would be unlabeled. If this initial glycogen content is different, the relative enrichment would be diluted and we would not know if this was due to flux differences or initial content difference. 

5. Lines 280-281: the authors state that “one possibility is that hepatic mitochondrial dysfunction partially impairs the ability to convert pyruvate or amino acids to new glucose”. There is no evidence to support this statement. The evidence described in the paper does not suggest that hepatic mitochondria suffer from disruptions in energy production. How can it be explained that mice that performed exercise for 30 min and struggle to use energy substrates in cardio-skeletal muscles do not present increased hepatic glucogenesis?

Response: Thank you. We have rephrased this sentence to indicate that it might be predicted that mitochondrial dysfunction would impair pyruvate and amino acid mediated gluconeogenesis. However, as the following sentences indicate, that is NOT what we saw. We have modified the text on lines 318-319 to make this clearer. 

6. Can the authors provide data on the concentration of muscle lactate and glucose? Additionally, while Fig 1B demonstrates that circulating lactate returned to WT levels after 30 minutes, it does not provide comparative data on the process of gluconeogenesis for muscle and blood. This comparison would provide valuable insight into the metabolism of glycogen and glucose in both tissues.

Response: Unfortunately, the tissues were not collected in a way that would allow for accurate measurements of lactate and glucose in muscle. The ischemia that happens immediately after sacrifice markedly increases muscle tissue lactate concentrations. While this is an interesting question for future investigation, the purpose of this study was to evaluate the effects of Tafazzin knockdown on hepatic glucose production and gluconeogenesis. We have commented on this in the Discussion (lines 374-377).

7. Have the authors conducted nutrient flux measurements at rest and during exercise using stable glycogen isotope tracer? These measurements would provide detailed information on the utilization and turnover of glycogen during exercise and at rest.

Response: We are not entirely clear what experiment the reviewer is suggesting here. To our knowledge, there is no commercial source of labeled glycogen and if there were, we are unsure how would an insoluble substance like glycogen be administered to mice. 

8. Have the authors investigated the expression levels of enzymes responsible for glycogenolysis to glucose, such as glucose phosphorylase, phosphoglucomutase, and glucose-6-phosphatase? This information would provide valuable insight into the underlying molecular mechanisms. Moreover, it would also be valuable to investigate the relative proportion of isotope labeled metabolites during both resting and exercise conditions in conjunction with their corresponding flux rates.

Response: Thank you for the suggestion. We now show that the expression of critical enzymes in glycogenolysis and gluconeogenesis (Pygl (glucose phosphorylase), G6pc (glucose 6 phosphatase) and Pck1 (phosphoenolpyruvate carboxykinase)) was not different between the two genotypes of mice immediately following the exercise test conducted at Vanderbilt. In addition, we assessed glycogen phosphorylase enzymatic activity in these livers and found again that it was not affected by genotypes. 

The reviewer is correct that for isotope tracing studies where a bolus of isotope is administered and isotope steady state is not possible, the enrichment of multiple metabolites in a metabolic pathway is very informative for nutrient fate and sourcing. For our studies we continuously infuse isotopes to achieve isotopic steady state. Under these conditions, the quantification of liver glucose fluxes (glycogenolysis, gluconeogenesis from glycerol and gluconeogenesis from TCA cycle intermediates) is dependent on evaluating both the proportional and positional isotope labeling in circulating glucose (which is predominantly derived from the liver). Importantly, the labeling pattern of different glucose precursors is “encoded” in the labeling of circulating glucose that we sample. This has notable advantages in that it prevents the need for and additional insight gained from metabolite labeling information beyond glucose. It also allows for the quantification of glucose fluxes over a time course in an individual mouse (i.e. multiple mice do not need to be sacrificed at each time point).

9. Have glucose tolerance tests or insulin resistance tests been conducted at rest and during exercise? Such tests would allow for the assessment of insulin and glucose uptake efficiency, as well as the effect of exercise on blood glucose levels.

Response: We thank the reviewer the suggestion. The objective of this study was to evaluate liver glucose and associated oxidative fluxes at rest and during exercise. Given this, we have not performed experiments to test glucose or insulin tolerance in these mice at rest or following exercise. It should also be noted that prior research from the laboratory of Dr. Hatch has shown that mice with long term Tafazzin knockdown exhibit lower blood glucose during glucose and insulin tolerance tests (PMID 27495222), suggesting that glucose disappearance is enhanced. Those studies were conducted in sedentary mice. 

Minor concerns:

1. Line 68-69, the use of “mature” and “immature” cardiolipin is not meaningful. Consider remodeled and unremodeled cardiolipin.

Response: Thank you. We have made this change. 

2. Figure 1a. should be more detailed to better indicate that this is the running regiment for each individual mouse.

Response: Since this seems confusing and we have additional data panels for this figure, we have now removed the treadmill protocol panel from the figure. The standardized protocol for all mice is indicated in the methods section. 

3. Figure 1B and 1C: ‘Plasma lactate’ is unclear. Does the y axis indicate post-exercise plasma lactate?

Response: As indicated in the figure legend, these values represent post-exercise glucose and lactate over time. 

4. Figure 3: The title of each figure should indicate that the glucose concentration was measured was during exercise.

Response: Corrected. 

5. Line 146: was total body weight recovered after surgery or just 10% recovered?

Response: We thank the reviewer for the opportunity to clarify our procedures. To ensure appropriate mouse welfare procedures during post-operative care and as a quality control procedure for our metabolic flux studies, we do not study mice that lose ≥10% of their pre-surgical body weight. This an extremely rare occurrence in our hands, however, it is important information for interpreting metabolic responses within and between groups in a study. We now clarify that the body weights of all study mice were within 10% of their pre-surgery body weight on page 8, lines 153-154. 

6. Line 15: What are MFA protocols?

Response: Metabolic flux analysis. We have now expanded this acronym to indicate the meaning. 

7. Last sentence of the second paragraph, lines 272-275: References should be cited with regard to “prior findings from our group and others support…”

Response: We have added references to this end. 

6. PLOS authors have the option to publish the peer review history of their article (what does this mean?). If published, this will include your full peer review and any attached files.

Do you want your identity to be public for this peer review? For information about this choice, including consent withdrawal, please see our Privacy Policy.

Reviewer #1: No

Reviewer #2: No

Reviewer #3: No

Reviewer #4: No

Reviewer #5: No

Reviewer #6: No

Additional comments from the academic editor:

1. Please, review and follow the PLOS ONE guidelines for financial disclosure, including whether the funders played any role in the study design, data collection and analysis, decision to publish, or preparation of the manuscript?

2. In the data availability statements, the authors states that "Data are available through Washington University. Please contact Brian Finck, PhD @ bfinck@wustl.edu.". However, authors should provide justifications for not providing the data undelying the study as supplementary file or data deposition in publicly available resources. Please, review and follow PLOS ONE guidelines on data availability. 

We look forward to receiving your revised manuscript.

---

## [Decision Letter · Decision Letter 1]

17 Aug 2023

Elevated Liver Glycogenolysis Mediates Higher Blood Glucose During Acute Exercise in Barth Syndrome

PONE-D-23-09366R1

Dear Dr. Cade,

We’re pleased to inform you that your manuscript has been judged scientifically suitable for publication and will be formally accepted for publication once it meets all outstanding technical requirements.

Kind regards,

Elsayed Abdelkreem, MD, PhD

Academic Editor

PLOS ONE

Additional Editor Comments (optional):

Reviewers' comments:

Reviewer's Responses to Questions

**Comments to the Author**

1. If the authors have adequately addressed your comments raised in a previous round of review and you feel that this manuscript is now acceptable for publication, you may indicate that here to bypass the “Comments to the Author” section, enter your conflict of interest statement in the “Confidential to Editor” section, and submit your "Accept" recommendation.

Reviewer #1: All comments have been addressed

Reviewer #2: All comments have been addressed

Reviewer #3: All comments have been addressed

Reviewer #4: (No Response)

Reviewer #5: All comments have been addressed

2. Is the manuscript technically sound, and do the data support the conclusions?

Reviewer #1: Yes

Reviewer #2: Yes

Reviewer #3: Yes

Reviewer #4: Yes

Reviewer #5: Yes

3. Has the statistical analysis been performed appropriately and rigorously? 

Reviewer #1: Yes

Reviewer #2: Yes

Reviewer #3: Yes

Reviewer #4: Yes

Reviewer #5: Yes

4. Have the authors made all data underlying the findings in their manuscript fully available?

Reviewer #1: Yes

Reviewer #2: Yes

Reviewer #3: Yes

Reviewer #4: Yes

Reviewer #5: Yes

5. Is the manuscript presented in an intelligible fashion and written in standard English?

Reviewer #1: Yes

Reviewer #2: Yes

Reviewer #3: Yes

Reviewer #4: Yes

Reviewer #5: Yes

6. Review Comments to the Author

Reviewer #1: The authors have addressed all of my queries and substantially improved the manuscript based upon the other referees comments. I believe the manuscript now provides an important contribution implicating that elevated anaerobic metabolism during rest and exercise are supported by higher hepatic glycogenolysis in the TazKD animal model.

Reviewer #2: LOS ONE PONE-D-23-09366R1 - [EMID:f7785e1000552902]

Elevated Liver Glycogenolysis Mediates Higher Blood Glucose During Acute Exercise in Barth Syndrome

Authors have responded sufficientely to previouösy raised objections

8/23

Reviewer #3: (No Response)

Reviewer #4: The authors have been responsive to the comments and questions raised in the initial review. The manuscript has been improved as a result, and provides an important contribution to the field.

Reviewer #5: All comments have been addressed in the Response to Reviewers', of which I note there were an astounding six (6) that participated. I question the journal's judicious use of Reviewer's time, and would like to note that had I known there were so many reviewers participating already, I would have declined this review request. I hope the journal reads this.

7. PLOS authors have the option to publish the peer review history of their article (what does this mean?). If published, this will include your full peer review and any attached files.

Reviewer #1: No

Reviewer #2: No

Reviewer #3: No

Reviewer #4: No

Reviewer #5: No

---

## [Editor Report · Acceptance letter]

22 Aug 2023

PONE-D-23-09366R1 

Elevated Liver Glycogenolysis Mediates Higher Blood Glucose During Acute Exercise in Barth Syndrome 

Dear Dr. Cade:

I'm pleased to inform you that your manuscript has been deemed suitable for publication in PLOS ONE. Congratulations! Your manuscript is now with our production department. 

Kind regards, 

on behalf of

Dr. Elsayed Abdelkreem 

Academic Editor

PLOS ONE